# Domain Modeling for Multi-Payload Planning of Experimental Satellite

## Peng Wu

Key Laboratory of Space Utilization, Technology and Engineering Center for space Utilization, Chinese Academy of Sciences
wupeng@csu.ac.cn

## Abstract

This paper introduce a domain modeling method for multi-payload planning of experimental satellite based on temporal planning paradigm, which may express temporal constraints and corresponding handlers in a universal style. This will allow user to model planning problem more efficiently, and guide planning process throw domain description files.

## Introduction

The experimental satellite usually carries multiple payloads including remote sensing, astronomy, scientific experiments to work alternately or at the same time. The payload may run on different working modes, and each working mode is a sequence of instructions.

Due to the nature of the experiment, scientists and engineers may have several requirements for the work of the payload, which may be summed up as the following aspects:

- Minimum/maximum duration constraint for single task
- Minimum/maximum interval constraint between two tasks
- Minimum/maximum interval constraint between two command instructions
- Task prerequisite constraint, that is, a task or an instruction must be executed before the constraint task
- Task mutual exclusion constraint, that is, a payload cannot work on two mode at the same time

In the process of satellite ground operation, the task constraint and its value may be adjusted frequently. Furthermore, if there is a task request that violates some constraint, the planner should processed according to the predefined conflict resolution method.

In fact, the formatted description of tasks, constraints, and their processing strategies is a domain modeling problem. Existing modeling languages such as PDDL and NDDL have been able to express such information well. But these languages are hard to use in engineering, due to our lack of support from domain-dependent planners.

Therefore, we proposed a general domain modeling method, which defined the expressions of tasks, events, constraints and conflict resolution strategies based on time interval algebra and time point algebra, and implements multi-payload task planning and scheduling based on greedy algorithms. This method conforms to the temporal planning paradigm described in Figure 1.

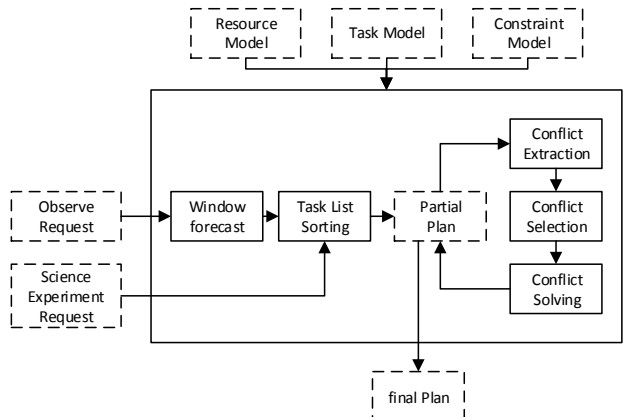

Figure 1: Temporal planning

## General Domain Modeling

Domain model contains three parts: resource, task and constraint.

### Resource model

Resource model is a triple:

- ResourceSet={R|R=(Name, Type, Capacity)}

The type can be continuous or discrete, specially if the type is discrete and the capacity is 1, this resource derive to bool type.

### Task model

The task model defines all the work modes that can be carry on by all instruments, and their command sequences. It has a hierarchical structure. The root is a payload set:

- PayloadSet = {Payload | Payload = (Name, WorkmodeSet)}

Each payload contains a WorkmodeSet which is defines as:

- WorkmodeSet={Workmode|Workmode=(Name, EventSet)}

Each workmode contains a EventSet which is defines as:

- EventSet={Event|Event=(Name, TimeRef, TimeShift, ParaSet, ResourceCostSet)}

In the definition of the event, the 'TimeRef' represents the time reference, which can only be T0 and T1 representing the start and end of the task respectively. The 'TimeShift'

represents the time offset relative to the time reference. The 'ParaSet' defines the device parameters to be adjusted for this event. The 'ResourceCostSet' represents all the resource requirement of this event, which is defined as:

- ResourceCostSet={ResourceCost|ResourceCost=(Name, Type , CostValue)}

Each 'ResourceCost' should specify the cost type, and the cost value. The type can be 'ByQuantity' or 'ByRatio' epresenting quantitative consumption and constant speed consumption respectively. Correspondingly, the 'CostValue' represents the amount of resources consumed at a time, or the rate at which resources are consumed.

## Constraint model

The constraint model consists of multiple time constraint. A time constraint CT consists of a set of constraint relationships and a set of constraint solvers.

- CT=(RelationSet, SolverSet)

The 'RelationSet' contains multiple constraint relationship groups RG. If one of the constraint relationship groups is satisfied, the constraint is considered to be satisfied. Each constraint relationship group RG contains multiple constraint relationships r. When all the constraint relationships r are satisfied, it is considered that the constraint relationship group RG is satisfied.

- RelationSet={ RG | RG=r1&r2& -----&rn }

the constraint relationship 'r' is a quad:

- r=(Nmaster, Nslave, type, value）

In the relationship r, the master node Nmaster and the slave node Nslave can be expressed as:

- N=(Payload, Workmode, Event)

The 'Payload', 'Workmode' and 'Event' are corresponding to the definition in the task model. They can also be 'Any', which represents arbitrary payload, arbitrary workmode, and arbitrary event, respectively, for the definition of fuzzy constraint.

The type of the relationship 'r' can be defined as:

- type∈{ Lower, Upper, PointQualitative, IntervalQualitative }

If the type is 'Lower' or 'Upper', the relationship is a quantitative point constraint. The 'value' is a const, and respectively represents:

$$T_{slave} - T_{master} > value$$
$$or\ T_{slave} - T_{master} < value$$

If the type is 'PointQualitative',

- value ∈ { before, ibefore, equal }

The 'value' can be 'before', 'ibefore' or 'equal', respectively represents:

$$T_{master} < T_{slave}$$
$$T_{master} > T_{slave}$$
$$T_{master} = T_{slave}$$

If the type is 'IntervalQualitative',

- value ∈ { b, bi, =, m, mi, o, oi, d, di, s, si, f, fi}

The 'value' respectively represents the 13 time interval relationships proposed by Allen (Allen 1983).

The constraint processor set 'SolverSet' represents a set of operations that can be used to solve conflicts when the task violates the constraint.

- SolverSet={SG|SG=S1&S2& -----&Sn }

The 'SolverSet' contains multiple solver groups SG, which are applied to solve the conflict, and only one SG is selected from the set. The choice is determined by the constraint processing algorithm. Each solver group SG contains multiple constraint solver S. If a solver group SG is applied, all the solvers S in the group should be executed.

There are many types of constraint solvers. The operations of the various types of solvers that have been defined so far are as follows:

- Si∈{Smerge,Sadd,Sremove,Sshift }

The meanings are:

- Smerge: merge two tasks
- Sadd：add a node(a workmode or a event)
- Sremove：delete a node
- Sshift：shift a node

The type of the constraint solver is not limited to the above, the solver type can be increased as needed, and the corresponding handler is added to the planner by means of a plug-in.

## Planning and Scheduling

There are many planners that use the temporal planning paradigm, such as EUROPA(Barreiro et al. 2012), CASPER (Chien et al. 2000), APSI (Fratini and Cesta 2012), etc.

In fact, the multi-payload planning problem does not require such a comprehensive planner. Instead, the planner should only handle task conflicts according to predefined rules. So we implement a planner based on the idea of plan-space planning method, that the planner should extract conflict, select conflict and then solve conflict continuously.

| **Function  ExtractConflict(TaskList, CTList)** |
| --- |
| **Input**: TaskList, CTList |
| **Output**: ConflictList |
| 1. **For** task1 **in** TaskList |
| 2.     **For** task2 **in** TaskList |
| 3.         **For** CT **in** CTList |
| 4.             **If** consistency(task1, task2, CT) == false |
| 5.                 ConflictList←CT(task1,task2) |
| 6. **Return** ConflictList |

First of all, we need a conflict extraction function. This function may extract all the conflict among the tasks input the function according to the CT List.

Then, the planning and scheduling algorithm may search through the Conflict List, and solve every conflict according to the solvers which are provided by the constraint model.

| **Algorithm PS(TaskList, CTList)** |
| --- |
| **Input**: TaskList, CTList |
| **Output**: Plan |
| 1.   Conflict←ExtractConflict(TaskList, CTList) |
| 2.   **While** Conflict≠∅ **do** |
| 3.      Sort(Conflict) |
| 4.      **For** CT **in** Conflict **do** |
| 5.         SolverGroup←Null |
| 6.         RemainConflictCount←∞ |
| 7.         **For** SG **in** SolverSet of CT(task1,task2) **do** |
| 8.            TrySolve(task1, task2, SG) |
| 9.            T←ExtractConflict([task1,task2],CTList) |
| 10.              **If** Length(T)<RemainConflictCount **then** |
| 11.                 RemainConflictCount←Length(T) |
| **12.**                 SolverGroup←SG |
| 13.         Solve(task1,task2, SolverGroup) |
| 14.         TaskList←Update(TaskList) |
| 15.      Conflict ←ExtractConflict(TaskList, CTList) |
| 16.   **Return** Plan←TaskList |

It is worth noting that the Solve(task1, task2, SG) function need the program entities corresponding to the solvers SG defined in the constraint model.

## Conclusion

Knowledge engineering for planning and scheduling plays a key role in planning system development. In this paper, we have presented a domain modeling method and provided a corresponding planning algorithm. It is a simplified application of temporal planning method, which allows users to easily define constraint models and corresponding conflict resolution operations.

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
