# OpenReview forum: "Domain Modeling for Multi-Payload Planning of Experimental Satellite"
_icaps-conference.org/ICAPS/2019/Workshop/KEPS — KEPS 2019_

### Official Review · AnonReviewer1 · 2019-05-08
**The paper lacks important information to be evaluated properly**

**Rating:** 2
**Confidence:** 2

**Review:**

It is difficult to evaluate the work proposed in this paper due to two reasons mainly. First, the lack of formal definitions of basic concepts, as for example, 'multipayload planning' or 'planning problem', makes hard to figure out the problems you want to solve and, more important, their complexity. Besides, concrete examples would facilitate the understanding. Second, there are no comparisons with other planning model as PDDL or NDDL, at least from the theoretical point of view, that provide insights into the advantages of using the modeling method proposed in the paper against other known planning-techniques. I also miss information on the role of the 'Resource Model', it is not even included in the displayed algorithms. I wonder whether you are assuming there are always enough resources to solve all the tasks, but even in that case, I think the cost of the solutions is important.  Probably, it has relation with the solvers, there are very little information about them and thus, it is unclear the characteristics or properties of the solution plans. For example, it is possible that a valid solution plan leaves unresolved tasks due to the 'Sremove' constraint solver?